# Differentiating Females with Rett Syndrome and Those with Multi-Comorbid Autism Spectrum Disorder Using Physiological Biomarkers: A Novel Approach

**DOI:** 10.3390/jcm9092842

**Published:** 2020-09-02

**Authors:** Nantia Iakovidou, Evamaria Lanzarini, Jatinder Singh, Federico Fiori, Paramala Santosh

**Affiliations:** 1Department of Child and Adolescent Psychiatry, Institute of Psychiatry, Psychology and Neuroscience, King’s College London, London SE5 8AF, UK; niakovid@gmail.com (N.I.); jatinder.singh@kcl.ac.uk (J.S.); federico.fiori@kcl.ac.uk (F.F.); 2Child and Adolescent Neuropsychiatry Unit, Infermi Hospital, 47923 Rimini, Italy; evamaria.lanzarini@gmail.com; 3Centre for Personalised Medicine in Rett Syndrome (CPMRS) & Centre for Interventional Paediatric Psychopharmacology and Rare Diseases (CIPPRD), South London and Maudsley NHS Foundation Trust, London SE5 8AZ, UK; 4HealthTracker Limited, 76–78 High Street Medical Dental, High Street, Gillingham, Kent ME7 1AY, UK

**Keywords:** Rett syndrome, autism spectrum disorder, children, machine learning, physiological biomarkers

## Abstract

This study explored the use of wearable sensor technology to investigate autonomic function in children with autism spectrum disorder (ASD) and Rett syndrome (RTT). We aimed to identify autonomic biomarkers that can correctly differentiate females with ASD and Rett Syndrome using an innovative methodology that applies machine learning approaches. Our findings suggest that we can predict (95%) the status of ASD/Rett. We conclude that physiological biomarkers may be able to assist in the differentiation between patients with RTT and ASD and could allow the development of timely therapeutic strategies.

## 1. Introduction

Rett syndrome (RTT) is a rare complex neurodevelopmental disorder that affects girls almost exclusively, about 1 in 10,000 female live births [1]. Children with RTT present with seemingly normal development in the first 6–18 months of life, followed by regression with loss of acquired purposeful hand movements and language skills and the appearance of hand stereotypies and gait abnormalities. The diagnosis relies on the clinical presentation only [2], even if up to 95% of patients with typical Rett syndrome and around 50–70% of atypical Rett syndrome harbour a mutation involving the methyl-CpG-binding protein 2 (*MECP2*) gene [3]. MECP2 is a critical epigenetic modulator that is crucial for the development of long and small genes involved in neuronal development, migration, and synaptic integration [4]. Although a diagnosis of RTT is based on the presence of at least two of the four main diagnostic criteria, and history of regression and several supporting minor criteria, the clinical phenotype is heterogeneous with regards to both the multi-systemic involvement and the severity of its presentation. The involvement of the autonomic nervous system (ANS) is pervasive, and a degree of autonomic dysregulation is present in the majority of RTT patients, to the extent that RTT can be considered a congenital form of dysautonomia [5]. Autonomic dysregulation manifests with a variety of emotional, behavioural, and autonomic symptoms, and it is linked with an increased risk of sudden death when compared to the general population [6,7]. Emotional, behavioural, and autonomic dysregulation (EBAD) is a term coined to describe the co-occurrence of dysautonomia in some children who display emotional and behavioural issues [8]. RTT patients with autonomic dysregulation typically present with signs of sympathetic over-activity such as anxiety/panic attacks, breathing dysrhythmia, temperature instability, peripheral vascular changes, gut dysmotility and cardiac abnormalities. Even if autonomic dysregulation tends to be a lifelong issue in RTT [7,9], few patients receive specific treatments for the autonomic dysregulation [10]. While certain mutations can offer some information on the clinical profiles of autonomic dysregulation in patients with RTT [10,11], in the majority of cases the genotype–phenotype relationship of autonomic dysregulation in RTT remains elusive. The prevalence of autonomic dysregulation in Rett patients is unclear [12], but the prevalence of symptoms caused by the inherent autonomic dysregulation can range from 68.8% for breath holding, 46.4% for hyperventilation, and 42.4% for abdominal bloating [10]. Moreover, up to 90% of Rett patients have a risk of developing epilepsy [13] due to the brainstem immaturity. The clinical management of these and other symptoms can have a profound impact on the inherent cardiorespiratory vulnerability in RTT patients and if not managed appropriately may increase the risk of sudden cardiac death in this patient group [7].

Autonomic dysregulation is not localised to patients with RTT. It can also present in children with complex neuropsychiatric profiles such as multi-comorbid autism spectrum disorder (ASD) and various organic diseases with a central nervous system involvement, such as acquired brain injuries and various rare diseases [14,15]. These patients are usually resistant to first-line pharmacological and non-pharmacological treatments. Clinicians often have to develop a personalised treatment strategy that takes into account all the single comorbidities presented by the patient, an essential step to convert them into treatment-responders. The use of wearable sensor devices might assist clinicians in delivering a personalised treatment strategy and in monitoring the response to pharmacological treatments. These devices can measure autonomic metrics that reflect changes to the autonomic nervous system status such as blood volume pulse (BVP), which is an indirect measure of heart rate (HR) and heart rate variability (HRV), and electrodermal activity (EDA) [16].

With the publication of the current version of the Diagnostic and Statistical Manual of Mental Disorders (DSM-5) [17], Rett syndrome was removed from the ‘Pervasive Developmental Disorder’ (PDD) umbrella, where it had been previously listed with ‘Autistic disorder’. This change was supported by the concept that Rett syndrome is a discrete neurological disorder, and ASD symptoms might not be salient in these patients and tend to be limited to specific stages of disease progression. However, a subset of patients with RTT can still present with ASD, and the diagnosis is made by adding the specifier ‘with a known genetic or medical condition’. RTT and ASD also have overlapping biological underpinnings such as disordered brain inflammation and neurotrophin signalling [18]. Moreover, some other evidence has shown that the visual fixation to social stimuli is characteristic to RTT patients when compared to those with ASD [19].

In this proof of concept exploratory study, the overarching goal was to evaluate whether machine learning approaches using physiological data captured using wearable sensor devices can assist in the identification of autonomic biomarkers that would be able to demarcate RTT patients from those with ASD with a high degree of accuracy.

## 2. Methods

Ten female patients with a clinical diagnosis of RTT and ten age-matched female patients with a diagnosis of autism spectrum disorder and intellectual disability (ID) were enrolled in this study. Given the exploratory nature of this study and because RTT is a rare disorder, a small sample size is justified. All the patients were part of the Centre for Interventional Paediatric Psychopharmacology and Rare Diseases (CIPPRD) and the Centre for Personalised Medicine in Rett Syndrome (CPMRS), National Specialist Child and Adolescent Mental Health Services based at the Maudsley Hospital, London (UK). Patients with RTT were recruited from the CPMRS and diagnosed according to the consensus clinical criteria as described previously [2]. The diagnosis of ASD was determined using the DSM-5 criteria [17] after clinical assessment undertaken by the multidisciplinary team (MDT) in the CIPPRD, using multi-source information obtained from face-to-face assessments, school information, and HealthTracker-based online completions of questionnaires such as the Profile Of Neuropsychiatric Symptoms (PONS) questionnaire [20]. Patients with neurometabolic or neurodegenerative conditions were excluded in order to avoid confounding factors related to the stage of the disease (Table 1). Informed consent was obtained to participate in the study.

Recent technological advances have brought wearable bio-sensors (e.g., electrocardiogram [ECG] sensors, sweating-rate sensors, respiration-rate body sensors, etc.) [21,22] to the forefront of clinical decision making. In our case, we used Empatica E4 wristbands, which can be placed on wrists or ankles to measure autonomic metrics in patients with ASD and Rett syndrome. E4 is a wearable research device that captures real-time biometric physiological data and software for analysis. Empatica E4 is easy to use since it is the shape of a simple watch, without requiring intrusive wires and electrodes [23]. Besides, compared to other devices, Empatica E4 was found to be equally accurate [24]. Wearable sensors have previously been used in patients with RTT [14,16] and the E4 device has been used to monitor stress to a high accuracy [23]. Further, a recent evidence synthesis of the use of different sensors showed that among the sensors evaluated, Empatica devices including the E4 were suitable for assessing physiological monitoring in individuals with ASD [25]. E4 devices are highly reliable and are designed to gather high-quality data, including:Movement and/or 3-axis acceleration (Accelerometers (ACC) [g])Electrodermal Activity (EDA) in micro Siemens (µS)Blood Volume Pulse (BVP) in nanoWatt (nW)

We applied machine learning algorithms as implemented in the WEKA machine learning toolkit [26,27]. These algorithms aimed to train a machine learning model and use it to classify new participants by predicting their labels correctly (‘ASD’ vs. ‘Rett’). We tested the discrete datasets separately.

The evaluation metrics that were used were the following (Figure 1):(1)***Accuracy*:** The accuracy measure provides information about the percentage of participants that were classified correctly by the model, regardless of whether they are characterised as ASD or Rett.
(1)$$Accuracy=\frac{Correctly\;predicted\;risk\;level}{Total\;number\;of\;participants}$$(2)***Precision* (Specificity):** The precision measure offers information about how many participants the model classified with the correct diagnosis when predicting the label *X*, where *X* can be ‘ASD’ or ‘Rett’.
(2)$$Precision=\frac{Correctly\;predicted\;risk\;level\;X}{Total\;predictions\;of\;level\;X},\;X=\left(\mathrm{ASD},\mathrm{Rett}\right)$$(3)***Recall* (Sensitivity):** The recall measure offers information about how many participants out of all participants with label *X* were correctly classified by the model. Precision and recall are metrics that provide label-specific information.
(3)$$Recall=\frac{Correctly\;predicted\;risk\;level\;X}{Total\;participants\;of\;level\;X},\;X=\left(\mathrm{ASD},\mathrm{Rett}\right)$$(4)***F*1 *Score*:** The *F*1 Score combines the precision and recall measures, since precision and recall provide different information.
(4)$$F1\;Score=\frac{2*\left(Precision*Recall\right)}{Precision+Recall}$$

First, the data were recorded and stored using the Empatica E4 wristband device. Then they were transferred to a computer, where the rest of the analysis was performed. Specific filtering methods for each data type were applied, and also numerical features were extracted using statistical functions. The data corresponds to a recording period of approximately 50 min during the patients’ general interactions while awake.

### 2.1. Filtering and Feature Extraction

The Empatica E4 has a photoplethysmography (PPG) sensor that provides BVP data using a proprietary algorithm at a sampling frequency of 64 Hz. As the BVP data contain much extraneous noise due to movement and environmental disturbances, the BVP signal was filtered using winsorization [23]. Winsorization is a statistical method for limiting extreme values to reduce possibly spurious outliers. Values below the nth percentile were set to the nth percentile, and data above the 100-nth percentile was set to the 100-nth percentile (experimentally, n was set to 5). Each second (64 samples) of the filtered BVP signal was summarised by calculating the mean of these BVP values.

The EDA measures the constantly fluctuating changes in specific electrical properties at the surface of the skin that arise when the skin receives specific signals from the brain. The Empatica E4 device captures the changes in the electrical conductance measured as skin conductance responses (SCRs). These abrupt increases in the conductance of the skin are measured in microSiemens (μS) on the wrist by flowing a minuscule amount of current between two electrodes attached to the wristband at a frequency of 4 Hz. The minimum amplitude threshold of SCR was set at > 0.01 µS, and deflections below this threshold were not counted as SCRs. This phasic rise in conduction linking baseline and peak conductivity is measured as the amplitude [28,29]. The aforementioned high-pass filter was applied, and the data were down-sampled at 1Hz by computing the average EDA values.

In addition, the E4 device has a 3-axis accelerometer (ACC) that measures the continuous gravitational force (g) applied to each of the three spatial dimensions (*X*, *Y*, and *Z*). Along with the raw three-axis acceleration data, a moving average of ACC is provided, and 32 samples are captured per second. The acceleration data are summarised using the following method [30]:(5)sum+= max3(abs(buffX[i] − prevX), abs(buffY[i] − prevY), abs(buffZ[i] − prevZ))

Then, the output is filtered by applying the following equation:(6)avg=avg∗0.9+(sum/32)∗0.1

### 2.2. Model Learning and Prediction

For the model learning and the classification-prediction procedure, we used the classification machine learning algorithms as implemented in the WEKA machine-learning toolkit [31]. WEKA is a collection of machine learning algorithms for data mining tasks for data pre-processing, classification, regression, clustering, association rules, and visualization [31]. Using data mining and machine learning, we built models for the prediction of the class (patient grouping) based on its attributes [32] such as the ACC, EDA, and BVP data.

We experimented with the following top algorithms that represent five different categories [27]:1.Trees (algorithms that use decision trees):
J48: an algorithm for building a decision tree [33].
2.Rules (algorithms that apply decision rules):
PART: Class for generating a PART decision list. Uses separate-and-conquer. Builds a partial C4.5 decision tree in each iteration and makes the "best" leaf into a rule [34].
3.Lazy (algorithms that use lazy learning):
LWL: Locally weighted learning. Uses an instance-based algorithm to assign instance weights. Can do classification (e.g., using naive Bayes) or regression (e.g., using linear regression) [35].
4.Meta (algorithms that apply or combine multiple algorithms (ensemble methods)):
Bagging: an ensemble algorithm that learns base models on subsets of the training data with the purpose of reducing variance and avoiding overfitting [36].
5.Function (algorithms that estimate a function):
Simple logistic regression: Classifier for building linear logistic regression models, with simple regression functions as base learners [37]. Note that this is analogous to linear regression, except that the dependant variable is nominal (as it is in our case) and not a measurement.


We applied the WEKA Percentage split option [27]. First, the data are shuffled randomly based on a random seed number by using the Fisher-Yates shuffle algorithm [38], which generates a random permutation of a finite sequence. The first 66% of instances in the shuffled data were used for training, and the remaining instances were considered as a test set [27]. The above procedure was repeated for each participant, and the test results were collected and averaged over all seeds, providing information about the performance of the models on label prediction and data classification. We repeated the procedure for many different random seeds in total, and the results were statistically evaluated using the measures that were mentioned before.

### 2.3. Ethics Approval and Consent to Participate

For RTT patients, the ethical approval was given by the London-Bromley research ethics committee (REC reference: 15/LO/1772). For ASD patients, anonymised data were collected as part of routine clinical data collection at the Centre for Interventional Paediatric Psychopharmacology and Rare Diseases (CIPPRD), South London and Maudsley NHS Foundation Trust. All patients were registered as patients in the CIPPRD.

## 3. Results

Figure 1 depicts the average accuracy results for all participants. The average accuracy of all the five algorithms that were used in this study was about 95%, suggesting that all the algorithms can classify 95% of the participants correctly, which in turn means that the machine learning model has been adequately trained and learned. Table 2 shows the spread of physiological measures between ASD and Rett groups. ASD patients had greater dysregulation of electrodermal activity and Rett patients had greater heart rate variability. Dysregulation of motor activity was greater in the Rett patients’ group. However, not all patients had abnormality across the board, suggesting that in both groups, there was a spectrum of autonomic dysregulation. Figure 2 presents the average precision, recall, and F1 score values for ASD vs. Rett participants for EDA and BVP datasets. In all cases, the average recall values were higher than 0.93, and this implies that the algorithms correctly predicted and classified both the ‘ASD’ and the ‘Rett’ participants. The average precision values were greater than 0.92 for both ‘ASD’ and ‘Rett’ participants, which implies that the algorithms correctly predicted and classified all the patients. Additionally, in all cases, the average F1 score was again greater than 0.93. Finally, the average values of ACC, EDA, and BVP in ASD vs. Rett syndrome patients are presented in Figure 3.

## 4. Discussion

Using our machine learning approach, we were able to correctly predict the status of ASD/Rett patients with a high degree of accuracy (95%), which has clinical implications as this approach could potentially be used to distinguish ASD and Rett groups. We provide an innovative methodology that improves biomarker resolution in studying ASD and Rett patients with complex psychopathology.

As would be predicted, the motor movement pattern of limb stereotypy in Rett syndrome as measured via the accelerometer readings differs from ASD patients (Figure 3). The main finding of this study was the ability to accurately differentiate between ASD and Rett patients using their EDA and BVP profiles, despite the heterogeneity of the population groups.

The sample was recruited from a national centre that manages patients with genetic, neurodegenerative, and neurometabolic disorders with emotional, behavioural, and autonomic dysregulation (EBAD). For this reason, the patient sample presents with multiple comorbidities associated with their RTT, ASD, and ID diagnoses, such as anxiety, depression, cortical malformations, or specific genetic mutations. Moreover, many of them were prescribed various pharmacological treatments at the time of the assessment. However, despite the clinical heterogeneity of both groups in the sample, the algorithm was able to correctly classify the patients in the RTT or ASD group with a high degree of accuracy. This finding underscores our premise regarding the specificity of BVP and EDA profiles in distinguishing RTT and ASD patients. However, in this study it is not possible to exclude an influence of specific comorbidities and what impact they have on the BVP and EDA values. Given that there is a high incidence of comorbid anxiety in individuals with ASD [39], anxiety could be responsible for the higher variability of EDA displayed by some of the individuals within the ASD group. On the other hand, EDA showed a minimal variability among all the individuals in the RTT group, possibly because of an impaired ability to anticipate stressful events and, therefore, a reduced cognitive component of anticipatory anxiety. The BVP values in the RTT group consistently reached higher peaks and lower troughs than the ASD group, suggesting abnormal autonomic modulation of the cardiovascular system. The features of the present study should also be viewed together with other work that has examined the autonomic profile in children with ASD [40]. Despite there being an overlap between the autonomic profiles in patients with RTT and ASD and a small sample size, our study was able to show that a relatively inexpensive and non-invasive wearable sensor was able to distinguish the autonomic profiles in RTT patients from those with ASD.

Further studies in a larger sample from community clinical settings will be required to validate and confirm the findings of the present study. The findings from the present study could lead to the development of an autonomic signature that might help in expediting the assessment and management of RTT before the prominent phenotypical symptoms emerge. Moreover, the identification of a biometric marker that is easy to implement and low cost will improve the utility of using such a tool in clinical settings where more sophisticated approaches such as the machine learning techniques described in this study are not available. At the same time, the findings observed in the ASD group can help to identify those deemed clinically to be at high risk for physiological anxiety and assist in timely intervention.

## 5. Conclusions

Our proof of concept exploratory study supports the premise that data collected from wearable sensors can be analysed through a machine learning approach to predict ASD and Rett status with a high degree of accuracy. The autonomic differentiation paves the way for further studies to pinpoint physiological biomarkers that can assist in the early diagnosis of ASD and Rett syndrome. Although the present study is a pilot study and, therefore, limits the generalisability of the findings, the study does support the view that a machine learning approach can be used to reveal the neurophysiological underpinnings between RTT and ASD, especially in children where early diagnosis is much needed.

## Figures and Tables

**Figure 1 jcm-09-02842-f001:**
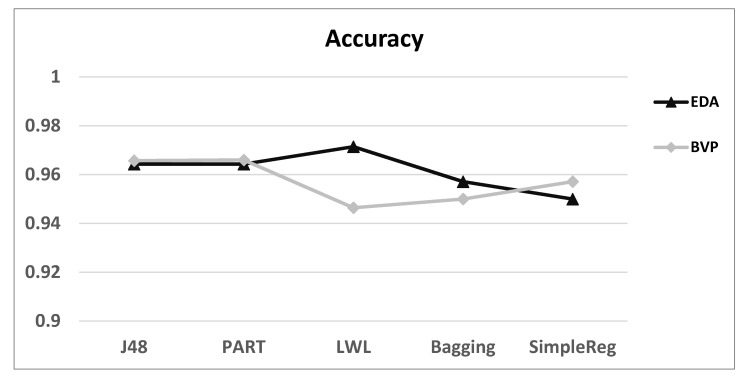
Average accuracy of predictions using the five algorithms: Results using EDA and BVP for correctly predicting ASD vs. Rett participant grouping across the total sample. Abbreviations and terminology: BVP (Blood Volume Pulse); EDA (Electrodermal Activity); J48 (algorithm for building a decision tree); LWL (Locally Weighted Learning); PART (Class for generating a PART decision list); SimpleReg (Simple Logistic Regression).

**Figure 2 jcm-09-02842-f002:**
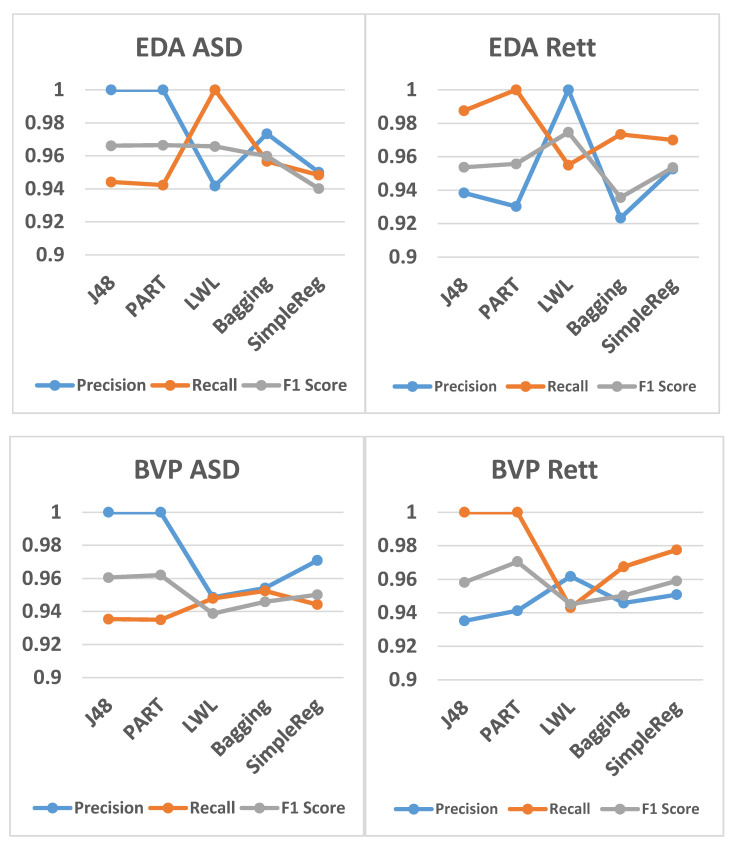
The 5 algorithms’ average precision (specificity), recall (sensitivity) and F1 score values for ASD vs. Rett participants when using EDA and BVP. Abbreviations and terminology: ASD (Autism Spectrum Disorder); BVP (Blood Volume Pulse); EDA (Electrodermal Activity); J48 (algorithm for building a decision tree); LWL (Locally Weighted Learning); PART (Class for generating a PART decision list); SimpleReg (Simple Logistic Regression).

**Figure 3 jcm-09-02842-f003:**
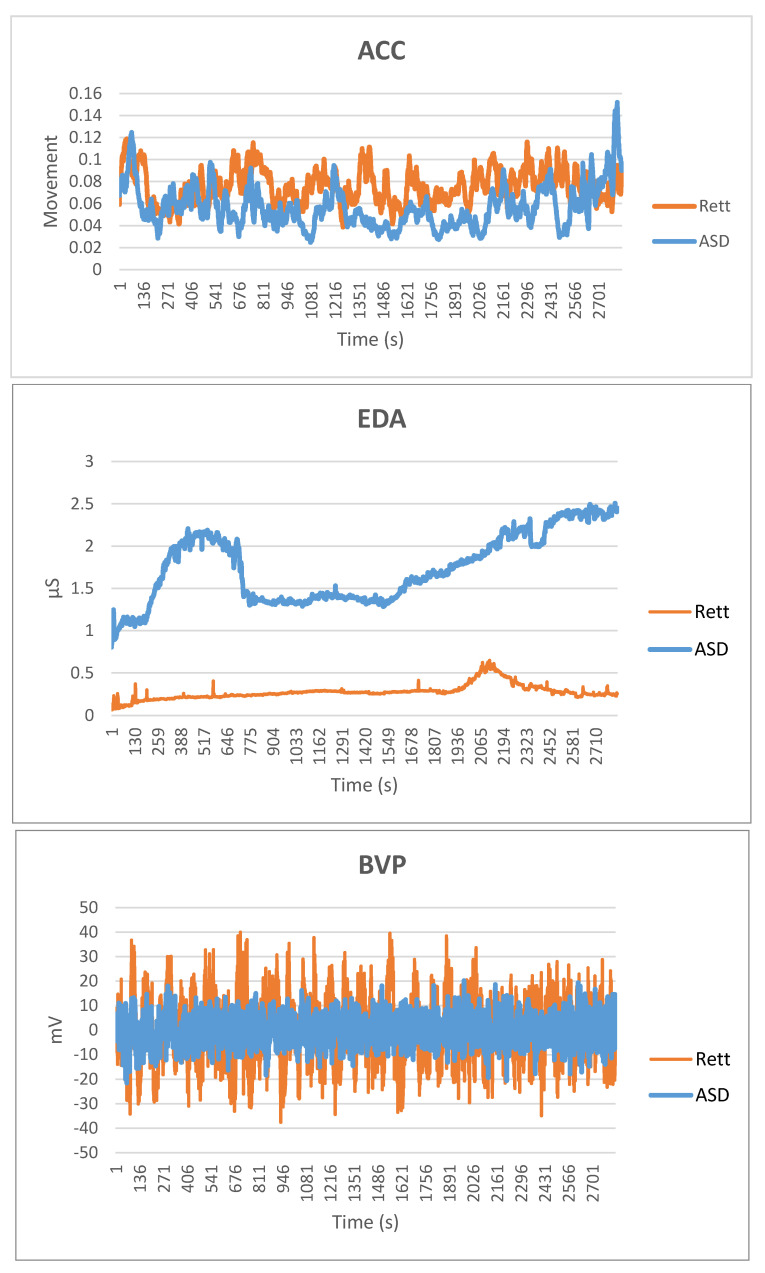
Average group values of ACC, EDA, and BVP in ASD vs. Rett syndrome patients. Abbreviations: ACC (Accelerometers); BVP (Blood Volume Pulse); EDA (Electrodermal Activity).

**Table 1 jcm-09-02842-t001:** Sample characteristics.

**ASD with ID Participants**				
**Patient No.**	**Age**	**Gender**	**Diagnoses**	**Medications**
1	19	F	Autism Spectrum Disorder Intellectual Disability Obsessive Compulsive Disorder Panic Disorder Generalised Anxiety Selective Mutism Depressive Disorder *SOX2* mutation causing anophthalmia with visual impairment Hypogonadotropic hypogonadism	Sertraline Propranolol Vitamin D
2	9	F	Autism Spectrum Disorder Intellectual Disability ADHD Oppositional Defiant Disorder Microcephaly Bulbar palsy with unsafe swallowing Interstitial lung disease	Sertraline Omeprazole Ranitidine Melatonin Alimemazine
3	16	F	Autism Spectrum Disorder Intellectual Disability History of Anti-NMDA Receptor Encephalitis post immunisation 3q27 Chromosomal Deletion Syndrome Plagiocephaly	Aripiprazole
4	17	F	Autism Spectrum Disorder Intellectual Disability Generalized Anxiety Disorder Depression with Psychotic Symptoms	Clozapine Sodium valproate Aripiprazole
5	13	F	Autism Spectrum Disorder Intellectual Disability Oppositional Defiant Disorder Multiple Anxiety Disorder	Aripiprazole Buspirone
6	15	F	Autism Spectrum Disorder Intellectual Disability Attention Deficit Disorder Left-frontal cortical dysplasia Symptomatic focal epilepsy Mild-right hemiparesis Tourette’s syndrome	Carbamazepine Topiramate Clobazam Perampanel Melatonin
7	16	F	Autism Spectrum Disorder Intellectual Disability Generalized Anxiety Disorder Cerebral Palsy Epilepsy Gastroesophageal reflux and gastro-jejunal feeding Visual impairment Scoliosis Menorrhagia	Lansoprazole Levetiracetam Clobazam Lacosamide
8	11	F	Autism Spectrum Disorder Intellectual Disability Attention Deficit Hyperactivity Disorder Generalised Anxiety Disorder Depression Hypokalaemia Respiratory symptoms Peripheral vascular problems Chromosome 2q13 deletion + *HECW2* gene mutation Pituitary micro-adenoma	Risperidone Carbamazepine Sertraline Melatonin Omeprazole Spironolactone Potassium
9	13	F	Autism Spectrum Disorder Intellectual Disability Oppositional Defiant Disorder Dysthymia Generalised Anxiety Disorder Four limb spastic cerebral palsy Epilepsy	Sodium valproate Baclofen Trihexyphendyl Omeprazole Domperidone Risperidone Citalopram Propranolol
10	16	F	Autism Spectrum Disorder Intellectual Disability Attention Deficit Hyperactivity Disorder Oppositional Defiant Disorder Generalised Anxiety Disorder Separation Anxiety Disorder Depression Episode Pentasomy X Spina Bifida Occulta Hypothyroidism Asthma	Fluoxetine Buspirone Risperidone Aripiprazole Trihexyphenidyl Cholecalciferol Lactulose Lansoprazole Levothyroxine
**Rett Participants**				
**Patient Nr.**	**Age**	**Gender**	**Diagnoses**	**Medications**
1	17	F	Rett Syndrome (*MECP2+)* Intellectual Disability Depressive Disorder Generalized Anxiety Disorder Gastro-oesophageal reflux Breath holding episodes	Sertraline Gaviscon Omeprazole Lactulose
2	5	F	Rett syndrome (*MECP2* C.473C) Intellectual Disability Generalized Anxiety Disorder Apneustic breathing pattern Gastro-oesophageal reflux	Melatonin
3	18	F	Rett syndrome Gastro-oesophageal reflux Scoliosis Epilepsy Urinary retention Impaired gut motility SIADH	Sodium valproate Lamotrigine Clobazam Azithromycin Buspirone Trihexyphenidyl Baclofen Ondasetron Ranitidine Sodium picosulfate
4	8	F	Rett syndrome Intellectual Disability Generalized Anxiety Disorder Gastrointestinal problems with constipation Mild scoliosis Bladder problems (urinary retention)	Osmotic laxatives
5	9	F	Rett syndrome (*MECP2* gene mutation-heterozygous for c.422A > C in exon 4) Intellectual Disability Epilepsy - absence episodes Hyperphagia and overweight Breathing irregularities	Sodium valproate
6	20	F	Rett syndrome Intellectual Disability Generalized Anxiety Disorder Depression Epilepsy Scoliosis Pulmonary Valve Stenosis Ataxic Gait	Sodium valproate
7	9	F	Rett syndrome Intellectual Disability Generalized Anxiety Disorder Respiratory Problems Epilepsy Gastro-oesophageal Reflux Scoliosis	Ranitidine Melatonin Osmotic laxatives
8	6	F	Rett syndrome (Heterozygous for a deletion of exon 3 and part of exon 4 of the *MECP2* gene) Intellectual Disability Generalized Anxiety Disorder Scoliosis and Joint Hypermobility Food intolerance Low weight	None
9	15	F	Rett syndrome Intellectual Disability Generalized Anxiety Disorder Disordered breathing with hyperventilation and apnoea Epilepsy PEG feeding and anti-reflux therapy Constipation and Bloating Scoliosis with spinal fusion Mild prolongation of QT interval	Omeprazole Melatonin Lamotrigine Diazepam Clobazam Pizotifen
10	14	F	Rett syndrome Intellectual Disability Generalized Anxiety Disorder Epilepsy Constipation Scoliosis	Movicol Vitamin D

Abbreviations: ADHD (Attention Deficit Hyperactivity Disorder); ASD (Autism Spectrum Disorder); ID (Intellectual Disability); HECW2 (HECT, C2, and WW domain containing E3 ubiquitin protein ligase 2); MECP2 (methyl-CpG binding protein 2); NMDA (N-Methyl- d-Aspartate); PEG (Percutaneous Endoscopic Gastrostomy); QT (Q and T waves on electrocardiogram); SIADH (Syndrome of Inappropriate Antidiuretic Hormone Secretion); SOX2 (Sex determining region Y-box 2).

**Table 2 jcm-09-02842-t002:** Colour coded table showing the physiological profile for ASD and Rett syndrome participants (Blue colour indicates normal values and red colour indicates abnormal values).

**ASD Participants**			
	**Activity Levels**	**Autonomic Function**	
Cases	ACC	EDA	BVP
1			
2			
3			
4			
5			
6			
7			
8			
9			
10			
**Rett Participants**			
	**Activity Levels**	**Autonomic Function**	
Cases	ACC	EDA	BVP
1			
2			
3			
4			
5			
6			
7			
8			
9			
10			

Abbreviations: ASD (Autism Spectrum Disorder); ACC (Accelerometers); BVP (Blood Volume Pulse); EDA (Electrodermal Activity).

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
