# Peer review of "Differentiating Females with Rett Syndrome and Those with Multi-Comorbid Autism Spectrum Disorder Using Physiological Biomarkers: A Novel Approach"

_jcm, 2020, doi:10.3390/jcm9092842_

Round 1

Reviewer 1 Report

The authors present a proof of concept exploratory study to determine whether machine learning approaches using physiological data from wearable sensors can help distinguish girls with RTT from those with autism-spectrum disorder.

Introduction: It would be helpful to explicitly state the proportion of RTT patients who can be expected to exhibit autonomic dysregulation.  Intro states a “majority” but further precision supported by the literature would be beneficial to your rationale.

Methods:

ASD and RTT are obviously difficult to diagnose with precision.  Given the importance of precision of diagnosis to the machine learning algorithms, the paper would benefit from greater elaboration on how diagnoses were made.  Some researchers require "research reliable" levels of agreement for diagnoses while others accept any clinically made diagnosis.  It would be useful to know more about how diagnosis was achieved (even for a proof of concept paper).

In describing the E4 device, there is a vague statement that it has been found to be equally accurate compared to other devices. It would be helpful to elaborate on this to better establish equivalence with other devices used and on what variables/measures so readers can make more judgment about the quality of the measurement.

Results are clearly presented and the use of multiple figures aids in the understanding of the results.  The discussion is balanced and acknowledges the exploratory, pilot nature of the paper and supports further investigation as indicated.

Author Response

Please see the attachment "Cover letter and response to reviewers_25Aug2020".

Reviewer 2 Report

Nice and interesting work. I onl suggest the authors to add some infromation concerning the neurophysiologic underpinnings and the potential generalizability of their findings. 

Author Response

(The authors gave the same response as above.)
